# Autophagy, Oxidative Stress and Cancer Development

**DOI:** 10.3390/cancers14071637

**Published:** 2022-03-23

**Authors:** Elisabeth Taucher, Iurii Mykoliuk, Melanie Fediuk, Freyja-Maria Smolle-Juettner

**Affiliations:** 1Division of Pulmonology, Department of Internal Medicine, Medical University Graz, 8036 Graz, Austria; 2Division of Thoracic Surgery, Department of Surgery, Medical University Graz, 8036 Graz, Austria; iurii.mykoliuk@medunigraz.at (I.M.); melanie.fediuk@medunigraz.at (M.F.); freyja.smolle@medunigraz.at (F.-M.S.-J.)

**Keywords:** oxidative stress, autophagy, cancer

## Abstract

**Simple Summary:**

Autophagy, as an important cellular repair mechanism, is important for the prevention of several diseases, including metabolic and neurologic disorders, and cancer. Hence, dysfunctional autophagy has been linked to these diseases, and in recent years researchers have tried to outline therapeutic targets in autophagy-related pathways as a treatment. With this review of the literature, we want to give an overview about the connection between oxidative stress, autophagy and cancer.

**Abstract:**

Autophagy is an important cellular repair mechanism, aiming at sequestering misfolded and dysfunctional proteins and damaged cell organelles. Dysfunctions in the autophagy process have been linked to several diseases, like infectious and neurodegenerative diseases, type II diabetes mellitus and cancer. Living organisms are constantly subjected to some degree of oxidative stress, mainly induced by reactive oxygen and nitrogen species. It has been shown that autophagy is readily induced by reactive oxygen species (ROS) upon nutrient deprivation. In recent years, research has increasingly focused on outlining novel therapeutic targets related to the autophagy process. With this review of the literature, we want to give an overview about the link between autophagy, oxidative stress and carcinogenesis.

## 1. Introduction

Autophagy is defined as the intracellular lysosomal degradation and recycling of cell organelles and misfolded proteins, and is tightly regulated by autophagy-related genes [1]. Nutrient supply (via mammalian target of rapamycin mTOR), energy availability (via AMP-activated protein kinase AMPK) and stress (via hypoxia-inducible factors HIFs) determine the function and activation of autophagy-related genes [2]. When the autophagy pathway is initiated, double membrane vesicles are formed, called autophagosomes. They scavenge damaged cytoplasmic cargo, organelles, lipids and glycogen which are marked with ubiquitin and readily detected by autophagy receptors [3].

Generally, autophagy is highly organized, sequestering misfolded and dysfunctional proteins that have exceeded their lifespan, as well as mutated proteins. After sequestering by autophagosomes, they coalesce to lysosomes, allowing for the degradation of sequestered particles [4,5]. The potential of autophagy to recycle cell detritus and to regulate cell homeostasis has been conserved from yeast to humans. Dysfunctional autophagy has been linked to a variety of diseases, e.g., infectious and neurodegenerative diseases, type II diabetes mellitus and malignant diseases [6,7,8,9]. Autophagy is subdivided into nonselective and selective autophagy [10]. Nonselective autophagy occurs when cells degrade the content of their cytoplasm in a large quantity. Selective autophagy means the targeting of specific organelles or proteins [11]. In recent years, research has increasingly focused on autophagy as a potential therapeutic target for several diseases, including cancer. Autophagy in mammals can also be subdivided into three categories, depending on the initiation mode of autophagy, i.e., macroautophagy, chaperone-mediated- and microautophagy. Figure 1 illustrates the three types of autophagy that have been described in mammals (Figure 1).

Human cells are constantly subjected to exogenously or endogenously produced highly reactive oxidizing compounds [13]. Among them are radicals and nonradicals (for instance, H_2_O_2_). What all oxidizing molecules have in common is the ability to take electrons from molecules upon close contact, ultimately damaging the cell structure [13]. Reactive oxygen species (ROS) and reactive nitrogen species (RNS) are most relevant in living organisms, being the main causes of oxidative stress. Most ROS occurring in the human body are derived from the mitochondrial respiratory chain, because mitochondrial complexes frequently leak electrons [14]. About 1 billion ROS molecules are generated intracellularly each day in humans, so there is a considerable physiological ROS flux which humans are chronically exposed to [15]. A highly efficient antioxidant capacity is therefore vital to counteract DNA damage inflicted by endogenous and exogenous (e.g., UV radiation-induced) ROS [16]. Among the most important defense mechanisms against oxidative cell damage are the antioxidant enzymes SOD, catalase and glutathione peroxidases [13,17]. These enzymes have the ability to scavenge ROS and even repair the reduced proteins or lipids [18].

It has been shown that autophagy is readily induced by ROS upon nutrient deprivation [19]. Autophagy initiates the building of an isolation cell membrane, termed phagophore [20]. The phagophore has the potential to recruit autophagy-related proteins and scavenge damaged cell organelles or proteins with an excessive life span. Autolysosomes are then built, eroding the phagophores’ contents, and recycling amino acids, fatty acids as well as nucleotides [21]. During the process of cancer formation, and particularly upon treatment with chemotherapeutics, autophagy is enhanced, which may result in cancer drug resistance and disease recurrence [22]. ROS have been shown to induce autophagy as a consequence of nutrient deprivation [19]. As proposed by an in-depth investigation of ROS in the context of nutrient deprivation, O_2_^−^ is the most important ROS in the process of autophagy, and is induced rapidly by the deprivation of glucose, glutamine or pyruvate [23]. A second important ROS molecule in autophagy is H_2_O_2_, with levels immediately rising as a consequence of starvation [24]. Moreover, the current evidence indicates that treatment with antioxidants partially or completely reverses autophagy [25].

Oxidative stress, as a DNA-damaging condition, effectively increases the mutation rate within cells and, over a longer time period, leads to tumor development [26,27]. Additionally, regarding the promotion of genomic instability, ROS have been found to specifically initiate pro-carcinogenic signaling pathways and contribute to carcinogenesis also by interfering with cellular proliferation, angiogenesis and metastatic dissemination [28]. Outlining the link between autophagy, oxidative stress and cancer, with a focus on therapeutic implications, is the aim of this review article. The different stages of the autophagy pathway are illustrated in Figure 2.

## 2. The Link between Endoplasmic Reticulum (ER) Stress, Oxidative Stress, Autophagy and Tumor Initiation

### 2.1. ER Stress and Autophagy

The endoplasmic reticulum (ER) is a cell organelle, consisting of a vast membrane-like structure in the cytoplasm. The ER has multiple functions, including the folding of newly synthesized proteins, keeping up the cell homeostasis and the synthesis of phospholipids, as well as the regulation of intracellular signaling cascades [30,31,32]. The rough ER holds the nuclear envelope domain, as opposed to the ER domain responsible for the synthesis of ribosomes [33]. The main functions of the ER are the transport and integration of proteins, lipid biosynthesis and the maintenance of calcium homeostasis [34,35]. Eukaryotic cells have the ability to rapidly adapt to a dysfunction of the ER through the activation of adaptive signaling cascades, which is termed ER stress [30]. The unfolded protein reaction (UPR) is one of these adaptive signaling pathways [36]. The UPR process primarily aims at attenuating the protein synthesis and recovery by modulating the cascade of the ER-associated degradation system (ERAD) which encodes chaperone proteins [37]. The activation of the UPR can trigger changes in the intracellular mitochondrial function or autophagy. When protein synthesis is dysfunctional, polypeptides fold and unfold in the lumen of the ER, leading to an accumulation of misfolded proteins [38]. The UPR response is a self-regulatory cellular defense mechanism to protect cells from irreversible damage [39]. The cancer microenvironment is generally hypoxic, low in glucose and abnormally vascularized [40]. The UPR reaction is therefore facilitated in the tumor microenvironment, and closely modulates apoptosis [41,42]. Cancer cells have the ability to upregulate the UPR signaling pathway, whereas healthy tissue surrounding the tumor does not [43].

When cells face harsh conditions, like nutrient deprivation, a disbalance in calcium homeostasis, toxins or oxidative stress, ER stress is initiated as one of the cell’s self-protection mechanisms [44]. In recent years, it was shown that ER stress can rebuild the normal function of the ER, and even the physiologic function of the whole cell, by means of autophagy. Autophagy, however, has two opposing features in cell metabolism: upon starvation, cells can degrade intracellular substances through autophagy to promote cell survival. However, when stimulating factors persist, and autophagy is executed at increasing rates, cell damage is aggravated, ultimately resulting in cell death [45].

The UPR can alleviate ER stress, helping the misfolded and unfolded proteins reconstitute their physiologic structure and activating cell autophagy, allowing for the homeostasis of the ER itself [46]. There are three signaling pathways triggered by ER stress, all involved in the activation of autophagy: first, the PERK-eIF2α pathway; second, the C/EBP-β-DAPK1-Beclin-1 pathway; and third, the IRE1–TRAF2–JNK pathway. Since ER stress-induced autophagy is one of the cell survival and self-protection mechanisms, it may explain resistance to antitumor therapy [46]. ER stress-induced autophagy is closely related to the PERK-eIF2α cell signal transduction pathway, which was shown in murine experiments: the gene knockout of PERK-inhibited cell autophagy was induced by the PERK-eIF2α pathway and strikingly decreased the number of autophagosomes in the cytoplasm in mouse embryonic tumor cells [47]. Furthermore, phosphorylated eIF2α mimicked cell autophagy, also highlighting the important role of eIF2α in ER stress-induced cell autophagy [48]. Another link between ER stress and autophagy is the increased expression of the transcriptional factor C/EBP-β upon ER stress, regulating the expression of DAPK1 which phosphorylates Beclin-1, finally resulting in autophagy. Another pathway mediating ER stress-related autophagy is the IRE1–TRAF2–JNK pathway. When IRE1 forms a complex with tumor necrosis factor receptor-associated factor-2 (TRAF-2), it may consecutively phosphorylate ASK1, which further phosphorylates JNK and induces autophagy [47,49,50].

As a conclusion, there is an evident relationship between ER stress and autophagy [44]. ER stress occurs as a consequence of strong intra- or extracellular stimuli. As a response mechanism, the UPR is activated to remove misfolded proteins that have built up in the ER. However, when the cell stress, i.e., the stimulus of ER stress, persists, the UPR is unable to remove the unfolded proteins. Consequently, ER stress will induce autophagy with the aim of reducing the swelling of the ER due to an ever-increasing amount of protein debris. Ultimately, the ER can go back to its normal size, the pressure of protein accumulation is alleviated and the cell survives [44].

### 2.2. Implication of Fasting and Calorie Restriction in Autophagy

The autophagy process is important for cell-to-cell communication, the mediation of protein secretion and the regulation of tissue-resistant stem cells [51,52]. In recent years, the importance of autophagy in organism homeostasis was increasingly recognized, with the goal of outlining molecular targets in autophagy to treat various pathologies [53]. Accurate data on this topic were gathered with mouse models featuring dysfunctional autophagy [54]. Oscillations in the intra- and extracellular metabolism can induce changes in the autophagy machinery, aiming at a strict balance between anabolic and catabolic cellular pathways [3]. Nutritional changes can induce autophagy via certain molecular players like mTORC1 and AMPK [55]. The lysosomal disposal breaks down macromolecules into amino acids, glucose, nucleotides and free fatty acids. Therefore, autophagy is a coordinator of energetic stress and maintenance of an organism’s energetic balance, including the adipose tissue, liver and exocrine pancreas [3,56,57]. In addition, an intact autophagy machinery interferes with the extracellular metabolome, mediating the interplay between tissue types and their response to bioenergetic signaling [53]. One such example is the decrease in acetyl-CoA which leads to autophagy induction and blocks anabolic reactions by activating AMPK and by inhibiting mTORC1 [58]. Upon starvation, autophagy causes the release of the DBI/ACBP/acyl-CoA-binding protein, which inhibits autophagy in target cells as a feedback mechanism, together with an increase in lipogenesis and food intake [59].

Metabolic diseases include type II diabetes, obesity and non-alcoholic fatty liver disease, which are mainly driven by excessive calorie consumption [60]. Changes in the autophagy flux affect the pathogenesis and progression of these metabolic diseases [61]. Insufficient autophagy is one of the pathomechanisms that contribute to metabolic syndrome, as it was observed that several autophagy-associated genes, e.g., ATG7, ATG4b, BECN2 and TFEB are substantially dysregulated, posing a predisposing factor for obesity [62,63,64,65]. The aberrant expression of the above-listed genes was linked to metabolic syndrome, irrespective of a normal or obesogenic dietary regimen. The experimental settings of autophagy-induction or the antibody-mediated neutralization of DBI/ACBP were found to counteract the metabolic anomalies caused by systemic metabolic dysfunction. Thus, it has been proposed that autophagy-stimulating therapies may be promising treatment strategies against metabolic disorders [60]. Still, autophagy is a highly tissue-specific process; therefore, autophagy inhibition as a future therapeutic option should also be considered as tissue-specific (e.g., in the adipose tissue), to minimize side effects [66]. Moreover, metabolic syndrome is most likely the combined result of the various functions of autophagy in adipocyte differentiation, formation of excessive fat deposits in the liver, pancreatic β-cell dysfunction, central mechanisms of food intake regulation and the inflammatory response [67,68,69].

Autophagy can be induced by starvation, aiming at the recycling of intracellular components into metabolic pathways to sustain functional mitochondria and energy homeostasis [2]. Preclinical analyses have shown that dietary restriction extends the lifespan and reduces the likelihood of lifestyle-associated diseases like diabetes mellitus, cardiovascular disease and cancer [70,71]. Decreased blood glucose induced by fasting allows for the activation of stress resistance pathways that affect cell growth, energy metabolism and protect against inflammation and oxidative stress [72]. In cell culture, nutrient deprivation consistently induces autophagy as well [73]. Currently, caloric restriction and fasting regimens are investigated as additional therapy options in oncology [70]. Autophagy has also been recognized as a crucial mechanism for the prevention of neurodegenerative diseases. Neurons depend on the clearance of dysfunctional and misfolded proteins, and the accumulation of such protein detritus promotes diseases such as Alzheimer’s and Parkinson’s disease [74,75,76,77]. Alirezaei et al. demonstrated that fasting over a short period of time upregulated neuronal autophagy in mice [78]. This analysis was completed on C57BL/6J and GFP-LC3 (Tg/+) mice deprived of food, and an upregulation of autophagy was observed after 24 h of fasting, and strikingly augmented after 48 h of fasting in cortical neurons and Purkinje cells in the cerebellum. Additionally, a reduction of phosphor-S6RP in Purkinje cell bodies of food-deprived mice was found, which is also an indicator of enhanced autophagy [78]. In another mouse model, the effect of fasting on Alzheimer’s disease was investigated [79]. Fasting significantly increased the number, size and signal intensity of autophagosomes in neurons. Alzheimer’s disease mice were compared to healthy control mice, and it was found that in the Alzheimer’s disease mice, autophagosome parameters were higher before fasting and increased more rapidly during the starvation period than in controls [79]. A mouse model on Charcot-Marie-Tooth disease showed that intermittent fasting for five months led to a significant increase in the autophagy-associated proteins ATG7 and Microtubule associated protein 1 light chain 3 (LC3). Hence, it is assumed that autophagy induced by fasting regimens may be a future therapeutic option in neurodegenerative diseases.

Liver autophagy has been investigated in-depth, because it is crucial for the maintenance of liver tissue homeostasis, and poses a protective mechanism against proteotoxicity, metabolic dysregulation, infectious liver conditions and liver cancer [80]. Rat studies have shown age-related lysosomal proteolysis in the liver tissue is reduced upon fasting or calorie restriction [81,82]. A similar report by Donati et al. showed chronic calorie restriction to stimulate autophagy in rats, which was demonstrated by an augmentation of autophagic proteolysis in rat liver cells. The maximum autophagy rates were observed in the calorie-restricted rats as compared to controls [83]. Kovacs et al. showed that fasting significantly increased the cytoplasmic volume fraction of autophagic vacuoles and of dense bodies, whereas insulin levels steadily decreased during fasting periods, as expected [84]. Another study showed that the number of lysosomes in rat liver cells significantly increased after 1–8 days of fasting [85]. The same study also showed a direct link between autophagy induction and starvation [85]. When the effect of calorie restriction on mitochondrial autophagy was investigated in mice, it was observed that the number of liver mitochondria was increased as a response to food deprivation, and autophagy was readily induced [86]. Fasting, and the consecutive autophagy induction, was also shown to be protective against damage by ischemia reperfusion in the liver of mice [87]. Even one day of fasting was sufficient to enhance autophagy and minimize ischemia reperfusion injury. SIRT1 activity was found to be upregulated following one day of fasting, because autophagic response due to starvation is regulated by SIRT1 [87].

Pietrocola et al. assessed the impact of starvation on the induction of autophagy in white blood cells [88]. Mice were divided into two groups, one with a standard diet and the other with a fasting regimen of 48 h. Additionally, nine healthy human subjects were enrolled into this study and fasted for four consecutive days (no-calorie diet with water, tea and coffee ad libitum). A striking increase in the number of microtubule-associated protein light chain 3 (LC3)B+ puncta per cell was observed in all white blood cells of mice in response to fasting, but only in neutrophils from human participants. These findings suggest that starvation increased the autophagy process across all white blood cell types in mice, but only in neutrophils from humans [88].

Summing up the above-listed data, fasting and calorie restrictions upregulate autophagy in humans. Physiological autophagy is crucial for cell homeostasis in mammals; therefore, fasting periods could be beneficial to some degree in the prevention and treatment of diseases where autophagy is usually dysregulated. Counterintuitively, it was also proposed to inhibit autophagy as a treatment option for metabolic disorders. More research is warranted to outline whether autophagy should rather be promoted or inhibited when treating metabolic syndrome and its related conditions. Moreover, the persistent upregulation of autophagy over an extended period of time is detrimental, which has to be considered when subjecting humans to fasting regimens.

### 2.3. Oxidative Stress and Autophagy

ROS are promoters of oxidative stress in mammals upon nutrient deprivation [19]. In macrophages, it has been shown that, when a bacterial infection occurs and ROS are generated, LC3 on phagosomes is recruited. The phagosomes, once modified by LC3, are then degraded via the autophagy process to prevent the escape of pathogens [89]. Many other studies have shown that in most cases mitochondria are the prime source of ROS as a prerequisite for autophagy induction [90,91]. The reason is that nutrient starvation leads to metabolic cell stress, increases the demand for ATP and consecutively leads to mitochondrial overburden [13]. As a result of mitochondrial overburden, electrons leak from the mitochondria which leads to an excessive production of ROS [13,92].

Mitochondria are the principal cell organelles for oxidative stress-mediated autophagy induction, because of their abundant ROS pool. Still, upon mitochondrial dysfunction, ROS are produced at excessive rates, and mitochondria are signaled to self-remove, which is termed mitophagy. This mechanism protects the cell from damage by oxidative stress [93]. In a recent study, it was demonstrated that mitochondria can also protect themselves against mitophagy [94]. Only when a cell is subjected to chronic nutrient deprivation, mitochondria depolarize and are fragmented, which is a last resort-mechanism when oxidative cell damage cannot be prohibited otherwise [95].

It has been demonstrated previously that treatment with antioxidants can prevent autophagy; hence, redox imbalance has a dichotomous role in autophagy. Autophagy induction generally happens fast upon excessive ROS production, meaning that the on/off-response dependent on ROS-sensitive proteins is rapid. AMPK is one such example, being activated upon H_2_O_2_ exposure [96]. Reduced glutathione (GSH) can induce autophagy when oxidized, even when no other autophagic stimuli are present [97]. This shows the importance of thiol redox homeostasis in autophagy mediation. In line with this assumption, there are several proteins involved in autophagy regulation, which are activated via cysteine residues. Examples would be the ubiquitin-like systems ATG7-ATG3 and ATG7-ATG10, and the phosphatase and tensin homologue (PTEN) [19].

The above-listed data demonstrate the close link between oxidative stress and autophagy induction. Thus, antioxidant treatment should also be considered as an autophagy-inhibiting treatment option in the future.

### 2.4. Tumor Necrosis Factor (TNF)-Induced Necroptosis and Autophagy

Conversely to the well-regulated and cytoprotective mechanism of autophagy, necroptosis constitutes an inflammatory form of cell death, leading to tissue necrosis [98,99]. Necroptosis is mainly mediated by death receptors like TNFRSF1A/TNFR1 (TNF receptor superfamily member 1A), although there are other pathways leading to necroptosis as well [100]. The necrosome, which is essentially an amyloid-like multiprotein complex, is formed after stimulation by trans-phosphorylated RIPK1 and RIPK3. After the CASP8 (caspase 8)-mediated cleavage of RIPK1 or RIPK3, the process of necroptosis, meaning the formation of necrosomes, can be inhibited [101,102].

Necroptosis as well as autophagy are mechanisms balancing cell death and survival, and a crosstalk between these two mechanisms has been reported [98]. Wu et al. demonstrated the link between autophagy and necroptosis, since RIPK3 connects directly with AMPK, phosphorylating its catalytic subunit PRKAA1/2 at T183/T172. After activation, AMPK phosphorylates the autophagy-mediating proteins ULK1 and BECN1. Moreover, Wu et al. could show that the lysosomal decomposition of autophagosomes is inhibited by TNF-induced necroptosis [98]. Therefore, RIPK3 and AMPK-activating kinases constitute a link between kinases regulating autophagy, and kinases regulating necroptosis.

TNF-mediated necroptosis has been linked to oxidative stress as well, because during necroptosis, ROS are generated, causing the peroxidation of lipids and increasing the permeability of mitochondrial membranes [103,104]. Although necroptosis is a vital mechanism keeping up a physiological balance between cell survival and cell death, it has been linked to various diseases as well, including cancer [105]. For instance, necroptosis was found to be impaired in chronic lymphocytic leukemia, and dysfunctional necroptosis was also linked to an increased risk of non-Hodgkin’s lymphoma [106,107]. Some studies have also linked necroptosis to metastasis. For example, the plant-derived substance shikonin, commonly used as an antiviral agent in Chinese medicine, reduced the lung metastasis rate in osteosarcomas by necroptosis induction [108]. One among several underlying mechanisms for metastasis inhibition by necroptosis is the accompanying abundant production of ROS [109]. Of note, metastatic cells usually escape apoptosis by restoring their ATP levels, restricting cellular ROS production and over-activating pro-survival signals (such as PI3K and Ras-ERK). However, the striking ROS burst resulting from necroptosis is one independent mechanism that successfully eliminates disseminated tumor cells [110].

The acknowledged apoptosis inducer TRAIL was shown to induce necroptosis instead of apoptosis upon an acidic extracellular pH in human HT29 colon and HepG2 liver carcinoma cell lines. The underlying mechanism for the conversion of TRAIL-induced apoptosis to necroptosis involved RIPK1/RIPK3-dependent PARP-1 activation [111]. In a mouse model, it was recently shown that murine prostate cancer cells were sensitized to TRAIL-mediated apoptosis after Map3k7 deletion [112]. Cell death in this investigation happened predominantly in the form of necroptosis, not apoptosis, due to an assembly of the necrosome linked to the autophagy process. The autophagy machinery, in this context, was induced by the p62/SQSTM1 recruitment of RIPK1. The authors of this study concluded, that in cancer, autophagy is a regulator of cell death, allowing for the switch between apoptosis and necroptosis [112]. Autophagy controls both apoptosis and necroptosis by serving as a framework rather than directly degrading cell detritus.

## 3. The Role of Autophagy in Cancer

Degenerative diseases, like cardiovascular or metabolic disorders, are suppressed by autophagy, whereas the role of autophagy in cancer is context dependent. Cancer initiation is suppressed by autophagy, because oxidative stress is minimized by preventing the accumulation of damaged cytoplasmic debris. Conversely, some cancers depend on autophagy for survival [2]. Autophagy is a promotor of survival in starvation, and similarly, malignant cells make use of autophagy to keep up their normal mitochondrial function and energy homeostasis. Therefore, autophagy inhibition is assumed to be a potential anticarcinogenic treatment strategy [1,113]. Originally, autophagy was thought to be a purely tumor-suppressive mechanism because essential autophagy-related genes like BECN1 (ATG6) feature allelic loss in human breast-, ovarian- and prostate carcinoma [114,115]. However, these results may have been confounded by a close proximity of autophagy-related genes to other tumor suppressor genes like BRCA1 on chromosome 17q21 [2,116]. Interestingly, it has been observed that a loss of the autophagy-related gene ATG5 promotes the formation of benign liver tumors in mice, which do not progress to carcinomas [117]. Similar studies have also shown that autophagy-related gene deletion is associated with benign tumor growth in the pancreas [118,119]. Therefore, it is assumed that autophagy suppresses tumor initiation in the liver and pancreas by the quality control of proteins and organelles, limiting tissue damage.

In tumor cells, autophagy has a versatile impact, mediating cancer initiation and maintenance, and ultimately, impacts treatment response [120]. Autophagy genes were found to be frequently mutated in cancers, or aberrantly activated. Cancer initiation can be linked to reduced autophagy levels, resulting in the accumulation of ROS and the activation of oncogenes [120]. Murine studies have shown that the deletion of the prime regulator of autophagy, Beclin-1, goes along with an increased likelihood of spontaneous tumor growth [121]. The monoallelic deletion of Beclin-1 was also observed in human breast-, prostate- and ovarian cancer. Conversely to cancer initiation, the autophagy machinery is frequently upregulated during cancer maintenance or in the stage of metastasis. Malignant tumors make use of autophagy, as a mechanism to better survive insufficient energy supply due to impaired perfusion. Notably, during anticarcinogenic treatments, an enhancement in autophagy is observed. Cancer stem cells are assumed to rely on upregulated autophagy as well [120,122]. A general increase in the risk of malignant diseases was previously linked to the monoallelic deletion of autophagy-related genes, such as ATG5, ATG7 or a total loss of ATG4C [117,123]. Autophagy therefore acts both as a tumor suppressor, and as a driver of cancer, depending on the stage of the malignant disease. Notably, a physiological constitutive level of autophagy is required for cell survival, because the knockout of autophagy-related genes, like Beclin-1, or AMBRA1, was embryonically lethal in mice [121,124].

It has been demonstrated that KRAS plays an important role in cancer autophagy, which also impacts treatment response [125]. RAS-driven tumors exert a particular metabolic flexibility, given their ability to take up and recycle metabolic intermediates from extracellular sources which serve them as fuel. Autophagy is one of these mechanisms [126], providing building blocks like amino acids, lipids and carbohydrates in order to maintain cancer cell survival. In KRAS mutant tumors, autophagy is specifically elevated [127] and sustained by increasing glycolytic rates, as well as sustaining mitochondrial respiration [128,129]. A research article by Kinsey et al. showed that protective autophagy by RAF-MEK-ERK inhibition constitutes a treatment approach for RAS-driven cancers [130]. The results from this study are in line with previous studies, also showing that autophagy is a response mechanism to RAS-RAF-MEK-ERK inhibition in cancer [131,132,133]. Kinsey et al. demonstrated that a combination of the MEK-inhibitor trametinib and the autophagy inhibitors chloroquine (CQ) and hydroxychloroquine (HCQ) promotes a regression of RAS-RAF-MEK-ERK-driven pancreatic ductal adenocarcinoma cells [130]. Thus, it becomes evident that upon RAS-MEK-ERK inhibition, the dependence of pancreatic (and presumably also other RAS-mutated) cancers on autophagy becomes much stronger.

### 3.1. Autophagy as a Novel Therapeutic Target in Cancer

Because of the evident close link between autophagy and cancer, indeed a pivotal role, targeting autophagy as a cancer treatment has become the focus of interest. Autophagy inhibition, based on findings from pre-clinical analyses, may improve the outcome of patients suffering from cancer [134]. In animal cancer models, autophagy inhibition was proven to lead to tumor reduction. Additionally, in vitro studies of genetically engineered mouse models and patient-derived xenograft mouse models showed that the treatment response to various anticarcinogenic agents was significantly improved upon autophagy inhibition [134,135,136].

CQ and the derivate agent HCQ are autophagy inhibitors that are already clinically approved. Their action mode lies in the deacidification of the lysosome, blocking the fusion of the autophagosomes with lysosomes and preventing cell debris degradation [137]. CQ may also improve the treatment response to other chemotherapeutic agents [138] and exerts anticarcinogenic effects which are not based on autophagy inhibition as well [139,140]. In a small study on patients suffering from glioblastomas (n = 18), the adjunction of CQ to their normal radio-chemotherapeutic treatment regimen resulted in a statistically significant prolongation of their overall survival, when compared to controls without CQ therapy [141]. In studies where the combination of CQ with radiotherapy as a treatment for brain metastasis was analyzed, the intracranial tumor control was found to be superior upon CQ [142,143]. The existing clinical data on autophagy inhibition as cancer treatment support the safety and at least some efficacy of these drugs. However, the underlying molecular mechanisms must be further elucidated, since autophagy inhibition could also have adverse effects [134].

Autophagy has been widely acknowledged as a survival mechanism in various types of cancer, protecting cells from undergoing programmed cell death [144,145,146,147]. Interestingly, the impact of autophagy on tumor cells is not always protective. It was demonstrated that within the same tumor sample, autophagy can promote or inhibit apoptosis, depending on the different cellular response mechanisms to apoptotic stimuli, such as CD95 ligand (CD95L) or tumor necrosis factor-related apoptosis-inducing ligand (TRAIL) [148]. The reason for this contradictory effect is the degradation of different pro- or anti-apoptotic regulators by the autophagy machinery [148]. An improved understanding of how precisely autophagy mediates a cell’s sensitivity to apoptotic stimuli is therefore warranted. Notably, increments in programmed tumor cell death are dependent on the stage of autophagy at the timepoint of inhibition. For instance, when inhibiting the maturation of autophagosomes in prostate cancer cells, decreased cell death was observed. However, in the same cell line, the inhibition of autophagosome turnover was a promoter of apoptosis [112]. In a study on autophagy inhibition in pancreatic ductal adenocarcinoma, it was shown that tumor cells are able to escape autophagy inhibition [149]. Pancreatic cancer cells upregulated and made use of the alternative pathway macropinocytosis to sustain their energy metabolism and survival. The authors propose that the switch from autophagy to macropinocytosis is conserved through evolution and may not be exclusive for cancer cells [149]. For the activation of macropinocytosis, the transcription factor NRF2 is needed, which is usually activated by tumor driver mutations, nutrient deprivation or oxidative stress [149]. NRF2 upregulation was also observed in a study on pancreatic cancer, resulting from KRAS-dependent metabolic reprogramming [150]. Upregulated NRF2 caused resistance to chemotherapy in pancreatic cancer and was linked to a poor prognosis. The underlying mechanism is most likely the rewiring of molecular pathways related to glutamine metabolism. The aberrant glutamine metabolism directly caused chemoresistance, and restrained the formation of stress granules, which is a hallmark of chemoresistance [150]. Both RAS and BRAF contribute to antioxidant processes in tumor cells, which are essentially pro-tumorigenic. One such antioxidant mechanism is the promotion of NRF2 expression, since NRF2 binds to antioxidant response elements and is linked to antioxidant gene expression [151,152]. Similar to the previously mentioned study, one investigation on Nrf2^−/−^ mice with pancreatic cancer revealed that these knockout mice had fewer pancreatic malignant precursor lesions when compared to Nrf2-expressing control mice [151]. By the adjunction of the antioxidant agent *N*-acetyl cysteine, the reduced proliferation of Nrf2-deficient pancreatic tumors is reversed, again indicating that the RAS-driven antioxidant program is essentially pro-carcinogenic and contributes to pancreatic cancer progression. Moreover, RAS also inhibited cancer cell apoptosis induced by H_2_O_2_ [153].

In another study on human colorectal cancer HCT116 cells, the potential of the autophagy inhibitor 3-Methyladenine (3-MA) to augment hypoxia-induced apoptosis was investigated [154]. Therefore, HCT116 cells were treated with 3-MA, hypoxia or 3-MA in addition to hypoxia. Treatment with hypoxia only led to an increased autophagy rate, whilst the combined 3-MA and hypoxia treatment significantly blocked hypoxia-induced autophagy while increasing hypoxia-induced cell apoptosis [154]. Thus, it is assumed that autophagy acts as a self-defense mechanism in hypoxia-treated colon cancer cells, which could be used as a future treatment option together with conventional chemotherapeutics.

Ferroptosis, an iron-related form of cell apoptosis, is defined as the accumulation of iron and cytotoxic lipid peroxides [155]. It was recently demonstrated that ferroptosis is related to autophagy through the degradation of ferritin, accompanied by an increment in intracellular iron and lipid ROS. The link between autophagy and ferroptosis may give rise to future anticarcinogenic treatment options [155]; however, more research is warranted to further elucidate this issue.

Several studies have been conducted on autophagy inhibition in pancreatic cancer, since a significant increase in autophagy has consistently been reported in pancreatic carcinoma tissues [156]. In pretreated patients with metastatic pancreatic cancer, HCQ was administered at a dosage of 400 mg or 600 mg twice daily. LC3-II in peripheral lymphocytes was used as an autophagy inhibition biomarker, but unfortunately no responses were seen in the patients and autophagy inhibition as measured by lymphocyte LC3-II levels was inconsistent [157]. In another study on pancreatic cancer and autophagy inhibition, the combination of CQ or HCQ with conventional chemotherapeutics was assessed [158]. Dose-limiting side effects were not observed, and three patients showed a partial response. However, no significant prolongation in progression-free or overall survival was observed when adding CQ or HCQ to chemotherapy [158]. Patients with resectable pancreatic cancer were treated with neoadjuvant gemcitabine together with HCQ in a Phase 1b/2 study [159]. A pathological sample analysis after the operation revealed no complete response following this treatment. The majority of patients showed a striking increase in LC3-II staining on peripheral blood mononuclear cells, as an indicator of autophagy inhibition. Patients who showed more than a 50% LC3-II increase had a better progression-free and overall survival as compared to those with less than 50% increase [159]. A study by Karasic et al. in which pancreatic cancer patients received standard chemotherapy with or without HCQ, revealed a significant increase in the response rate; still, no significant improvement in progression-free survival was observed [160]. In patients with HCQ treatment, more chemotherapy-related adverse effects were seen, accompanied by HCQ-related side effects like visual impairment and neuropsychiatric symptoms [160].

While previous research has mainly focused on autophagy inhibition as an anticarcinogenic treatment approach, there is still debate about whether activating autophagy could be a strategy as well. In preclinical models of pancreatic cancer, autophagy was activated by rapalogue-mediated mTOR inhibition, which resulted in some promising effects [157]. However, no translational effect was seen when activating autophagy as a treatment strategy in humans [161]. When applying rapalouges (i.e., rapamycin derivates) to achieve mTOR-mediated autophagy inhibition, the losses of the mTORC1 regulators TSC1 and TSC2 have been observed in clinical studies [162,163,164]; however, the direct correlation with the status of autophagy is still lacking. Notably, preclinical studies have even shown that the adjunction of mTOR inhibitors to autophagy inhibitors potentiate cytostatic effects [165]. This counterintuitive effect may be due to the induction of autophagy as a survival mechanism which is triggered by mTORC1 inhibition, and not a direct effect of mTORC1 on autophagy activation [156,165]. Moreover, nutrient deprivation alone is more effective at autophagy induction than rapalogues [6]. When applying rapalogues as single mTOR inhibiting agents, no activity was observed in patients suffering from pancreatic carcinomas [157]. Notably, although the roles of mTOR and AMPK in autophagy are established and well-investigated, there is still a missing link. Novel research indicates that phospholipase D, interconnecting with mTOR and AMK, could also serve as a potential target for cancer treatment [166]. According to a study performed by Jang et al. phospholipase D enhanced the autophagic flux via ATG1 (ULK1), ATG5 and ATG7, all three being important gene products of the autophagy machinery involved in autophagosome formation. Phospholipase D was also shown to suppress autophagy, interconnecting with both mTOR and AMPK, thereby altering the ULK1 phosphorylation pattern. When phospholipase D was inhibited, cancer regression was observed as a result of autophagy inhibition, suggesting phospholipase D inhibition as a new strategy to target autophagy in cancer [166].

Of note, when implementing autophagy inhibition in clinical practice, it must be considered that serious adverse effects were seen in autophagy-deficient (Atg5^−/−^) mice [54]. Toxicity included neurodegeneration, a high susceptibility to infection, an imbalance in glucose homeostasis, neurodegeneration, injury of muscle-, liver-, and pancreatic tissues and heart failure [167,168]. More research is clearly warranted to outline the toxic side effects of autophagy inhibition in humans and to exactly determine the risk–benefit ratio. Notably, in preventive settings, non-pharmacological autophagy-inducing measures, such as exercise and calorie restriction, have proven beneficial effects on general health [169,170].

### 3.2. Autophagy Biomarkers in Cancer

Changes in the rate of autophagy in cancers could be measured with biomarkers in the future. Thereby, it would be possible to determine which cancers feature a significant upregulation of autophagy, and when it would be feasible to target autophagy. One study showed that an increment in intra-tumoral HCQ was linked to an increase in LC3-II puncta formation and an increase in sequestosome 1, which determines the autophagosome turnover rate. This effect was not seen in control tumors without HCQ treatment [171]. Since HCQ evidently blocks autophagy in cancers, LC3-II and sequestosome 1 could be used as autophagy biomarkers. In peripheral blood mononuclear cells, the count of double membrane vesicles was evaluated by means of transmission electron microscopy (TEM) [134,172]. This approach was found to be insufficient for autophagy detection, because the number of double membrane vesicles does not correlate with autophagy inhibition in tumor samples measured by LC3-II and sequestosome 1 [172]. It was considered to measure HCQ in the blood plasma as an indicator of intra-tumoral HCQ. However, differences in HCQ abundance between tumors and plasma can be up to a 100-fold; thus, a plasma analysis is not a reliable approach [171].

Apart from HCQ, new autophagy inhibitors have been developed, like Lys05 and its derivate Lys01, which are more effective in accumulating in and deacidifying lysosomes, resulting in a more potent autophagy inhibition at lower doses. HCQ, by contrast, exerts the same potential as Lys05 and Lys01 only at toxic dosages [134,173]. The effects of Lys05 and Lys01 are measurable by LC3-II and by the increment in autophagosomes as verified by transmission electron microscopy [173].

Recent studies are now focusing on new autophagy biomarkers which also measure intra-tumoral hypoxia, establishing a correlation between hypoxia and autophagy. Functional imaging is performed via positron emission tomography/computed tomography (PET/CT) utilizing hypoxia tracers labeled with an 18F-fluorine isotope and [18F]-HX4 [18F-flortanidazole] [174]. Future studies have been designed, planning to assess the combined effect of the proteasome- and vorinostat-mediated inhibition of histone deacetylase (HDAC) on autophagy. In fact, it has previously been demonstrated that HDACs increase the rate of autophagy by gene transcription regulation [174]. Proteasome inhibitors were also shown to induce autophagy which is most likely a mechanism contributing to treatment resistance [175]. Table 1 shows the current trials aiming at investigating autophagy biomarkers in cancer (Table 1).

## 4. Conclusions

The above-reviewed data show a complex link between autophagy, oxidative stress and cancer. Autophagy is an important cell survival mechanism, and a functional autophagy machinery is crucial for tissue homeostasis. Oxidative stress, i.e., ROS mostly generated by mitochondria, induces autophagy. Nutrient deprivation often results in enhanced ROS production (via metabolic stress and an increased ATP demand resulting in mitochondrial overburden) and oxidative stress as well, but may also independently trigger autophagy as a cellular stress response mechanism.

When investigating the role of autophagy in cancer, current data are controversial. Autophagy inhibitors are already available in clinical practice and have so far shown limited effectiveness. It is generally acknowledged that cancer cells make use of autophagy to escape programmed cell death, so inhibiting autophagy is used as an anticarcinogenic treatment approach. However, it must be considered that autophagy-related gene knockout mice are unable to survive, and inhibiting autophagy may disrupt the homeostasis of healthy tissues. Notably, as a complementary treatment strategy in cancer, fasting and calorie restriction regimens have been successfully applied, aiming at enhancing rather than blocking autophagy.

In this respect, more research is clearly warranted to further clarify in which cases autophagy inhibition is the treatment of choice, and under which circumstances autophagy enhancement is preferable.

## Figures and Tables

**Figure 1 cancers-14-01637-f001:**
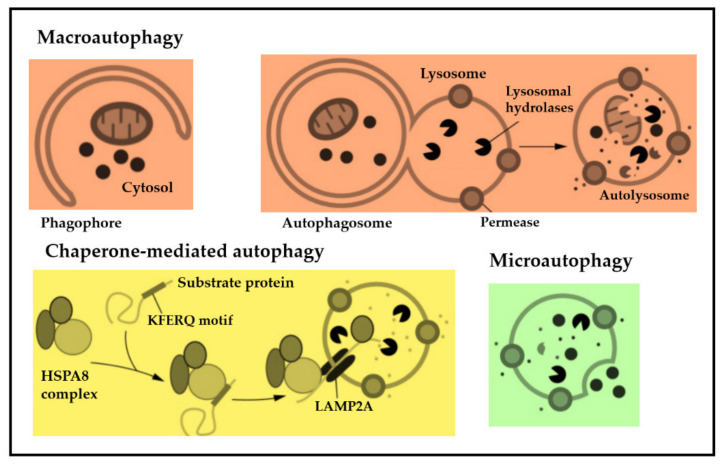
There are three types of autophagy in cells of mammals. In macroautophagy, double-membrane vesicles and autophagsosomes are created de novo in order to pack and transfer cell debris to the lysosome. In chaperone-mediated autophagy, misfolded proteins are carried individually across the lysosomal membrane. In microautophagy, cell detritus is taken up directly following the invagination of the lysosomal membrane. What all three autophagy modes have in common, is the complete breakdown of cell cargo and the secretion of degradation end products back into the cytosol for reuse. Figure adapted with kind permission from Parzych et al. [12].

**Figure 2 cancers-14-01637-f002:**
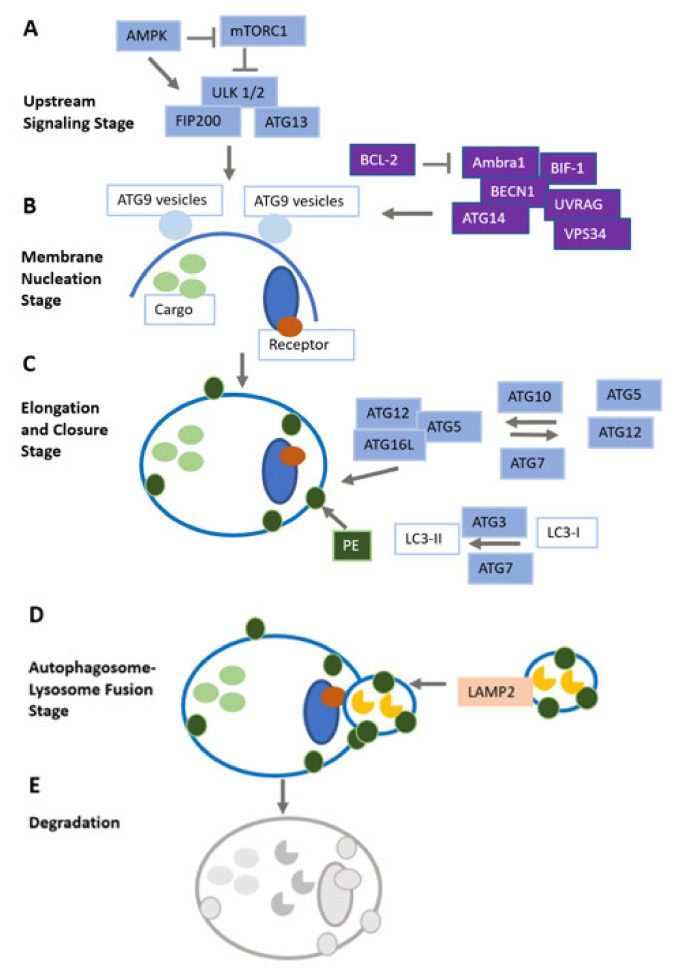
(**A**): Autophagy signals mostly converge at the level of mTOR protein complexes, which are regulated by AMPK. mTORC1 may limit autophagy by inactivating ULK1/2; (**B**): The class III phosphatidylinositol 3-kinase (PI3K) complex mediates the membrane nucleation stage and the initial building of the phagophore; (**C**): The elongation and closure stage depends on two ubiquitin-like conjugation systems. In the first system, ATG12 links to ATG5, mediated by ATG7 and ATG10. Next, ATG16L1 and ATG12-5 are recruited which results in the formation of a larger protein complex. Consecutively, lipid molecules are conjugated by ATG12-5-16L1 oligomers, serving as E3 ligases. Lipid-conjugated ATG8 is a prerequisite for the elongation and closure stage; (**D**): After autophagosomes fuse with late endosomes or lysosomes, they can exert their full lytic capacity; (**E**): Following the integration of the outer membranes of autophagosomes, cell cargo located in the inner membrane is degraded by lysosomal hydrolases. Figure adapted after Kocaturk et al. [29].

**Table 1 cancers-14-01637-t001:** Current trials on the investigation of autophagy biomarkers. Table adapted after Levy et al. [147].

Biomarkers of Autophagy	Assessment Mode	Tumor Entity	ID of Clinical Trial
Hypoxia	18F-EF5 PET LC3-II staining of tumor tissue Expression analysis of autophagy genes	Clear cell ovarian cancer	NCT01881451 [176]
Hypoxia	18F-HX4 PET Expression analysis of autophagy genes	Cervical cancer	NCT02233387 [177]
Autophagosome count	Number of autophagy vesicles in peripheral blood mononuclear cells	Multiple myeloma	NCT01594242 [178]
Aberrations in cell metabolism	18FDG PET Number of autophagy vesicles in peripheral blood mononuclear cells	Colorectal cancer	NCT01206530 [179]
Aberrations in cell metabolism	Analysis of metabolism-related serum parameters	Advanced-stage p53 mutated cancers	NCT02042989 [180]
Aberrations in cell metabolism	MRI study including MR spectroscopy and diffusion weighted imaging	Cervical cancer	NCT01874548 [181]

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
