# Peer review of "Autophagy, Oxidative Stress and Cancer Development"

_cancers, 2022, doi:10.3390/cancers14071637_

Round 1

Reviewer 1 Report

Tumor necrosis factor alpha (TNFα) has been reported to induce necroptosis and regulate autophagy, and there are many reports about. TNF-related apoptosis inducing ligand (TRAIL) regulates cancer cell death through necroptosis, associating the necrosome with the autophagy pathways, mediated by the p62/SQSTM1 recruitment of RIPK1. The authors should add a paragraph about the process to achieve a comprehensive and complete understanding of this other cell death pathway.

Author Response

We thank Reviewer 1 for the revision of our article. We agree that explaining the additional pathway (TRAIL-mediated pathway, linking autophagy and necroptosis) would be a feasible addition to this paper. Accordingly, we have now added a new chapter (chapter 2.4), where we describe the link between autophagy and necroptosis (lines 353 to 397).

All changes are written in red in our manuscript.

We hope that, after careful revision, Reviewer 1 now deems our manuscript appropriate for publication.

Reviewer 2 Report

A timely review article by Dr. Taucher elaborates the role of autophagy and oxidative stress in cancer development and therapeutics. This review is a very well-written and informative piece of work in the field of autophagy-cancer biology field. Though few things need to be addressed before it is ready for acceptance. They are as follows:

  1. It has been well established that KRAS plays a significant role in autophagy and this connection has significant therapeutic implications (PMID: 30833748 and PMCID: PMC8045781). Authors need to mention this aspect at least a few lines by referring to the relevant works as mentioned.
  2. The authors have discussed the role of mTOR and AMPK in autophagy throughout the text. But there is a missing link- PLD/Phospholipase D which interconnects mTOR and AMPK for their significant role in autophagy (PMID: 24317201). This aspect should be discussed which will give a more detailed view of autophagy in cancer development. 
  3. While discussing the role of NRF2 in macropinocytosis authors should discuss the role of NRF2 in stress granule formation and how tumor stress plays a role in cancer metabolism and therapeutics (PMID: 31911550 and PMID: 35147163).
  4. The authors should add another figure depicting all related signaling pathways which have been discussed in this review article. This will be useful as an illustrative overview of this review article.

Author Response

Ad 1) We thank Reviewer 2 for the careful revision of our manuscript. As a response to the rightful remark about mentioning KRAS in relation to autophagy in tumorigenesis, we have now added a new paragraph (lines 436 to 452), citing the two papers Reviewer 2 recommended.

Ad 2) We agree with this comment as well, and accordingly, we have added some information about phospholipase D from the recommended article by Jang et al. (lines 567 to 577).

Ad 3) As a response to this suggestion, to add some more information on NRF2, we have cited both articles as suggested and have added a few sentences on NRF2, stress granule formation and the link to RAS-driven carcinogenesis (lines 497 to 513).

Ad 4) We have added an additional figure (figure 2 on page 3), where the autophagy pathway is illustrated in more detail.

All changes are written in red in our manuscript.

We hope that, after careful revision, Reviewer 2 now deems our manuscript appropriate for publication.

Round 2

Reviewer 2 Report

All concerns have been addressed, ready for acceptance.